# Dramatic Declines of Evening Grosbeak Numbers at a Spring Migration Stop-Over Site

**W. Douglas Robinson** [1,*], **Jessica Greer** [1,2], **Juliana Masseloux** [1,3], **Tyler A. Hallman** [1,4] and **Jenna R. Curtis** [1,5]

1 Oak Creek Lab of Biology, Department of Fisheries, Wildlife, and Conservation Sciences, Oregon State University, Corvallis, OR 97331, USA; jessicagreer34@gmail.com (J.G.); jamasseloux@gmail.com (J.M.); tyler.hallman@vogelwarte.ch (T.A.H.); jennarcurtis@gmail.com (J.R.C.)
2 Department of Neurobiology, Physiology, and Behavior, University of California Davis, One Shields Avenue, Davis, CA 95616, USA
3 Zoological Society of London, Kanchanburi 71000, Thailand
4 Monitoring Department, Swiss Ornithological Institute, Seerose 1, 6204 Sempach, Switzerland
5 Cornell Lab of Ornithology, 159 Sapsucker Woods Road, Ithaca, NY 14850, USA
* Correspondence: douglas.robinson@oregonstate.edu

**Abstract:** Evening Grosbeak (*Coccothraustes vespertinus*) populations have been hypothesized to be in steep decline across North America. Data characterizing long-term changes are needed to quantify the magnitude of the declines. We surveyed grosbeaks at a spring migratory stop-over site in Corvallis, Oregon, USA, where birds gather annually during April and May to feast on elm (*Ulmus* spp.) seeds before departing to breeding sites. An estimate produced by a statistics professor in the 1970s indicated peak numbers were 150,000 to 250,000 birds. Our surveys in 2013–2015 found annually variable numbers, from a few hundred grosbeaks in the lowest year to less than five thousand birds in the highest year. If the original estimate is approximately true, Evening Grosbeak numbers have experienced dramatic declines, averaging −2.6%/year, over the last four decades. Our local observation of declines during spring aligns with declines documented in winter across North America by bird feeder studies and in summer by the Breeding Bird Survey. We explore potential explanations for the changes in population size, such as influences of spruce budworm outbreaks, disease, and decreased structural diversity of forests owing to harvest practices. We also consider the challenges of interpreting changes in abundance of species with exceptionally variable populations, especially if population fluctuations or cycles may have long periodicities. Finally, we call for additional planned surveys to track the numbers of this enigmatic and charismatic species.

**Keywords:** biodiversity benchmarks; *Coccothraustes vespertinus*; Evening Grosbeak; American elm; Dutch elm disease; migratory stopover; bird population decline; snapshot surveys; spruce budworm; *Ulmus americana*

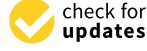

## 1. Introduction

Organismal abundance is an essential component of diversity [1]. As numbers fluctuate, patterns of diversity and species' interactions with the environment fluctuate as well [2]. Long-term directional changes in population sizes of individual species, particularly formerly abundant ones, can affect ecosystem-level processes more than the loss of rare species [3]. Thus, the documentation of decreasing population size and its potential causes can be informative. An example of apparently dramatic declines in the North American avifauna involves Evening Grosbeaks (*Coccothraustes vespertinus*).

Evening Grosbeak abundance has declined across its geographic range in North America. For example, between 1988 and 2006, Evening Grosbeaks disappeared from half of sites reporting visits to bird feeders, while flock sizes declined by 27% [4]. Data from the breeding season also indicated a range-wide decline of 4.3%/year with declines being steeper in the eastern than in the midwestern and western portions of the range [5].

Evening Grosbeaks are an irruptive species [6–8], breeding in coniferous forests of northern and western North America, then wandering episodically as their primary food sources, including spruce budworms and conifer seeds, fluctuate geographically and temporally in availability [9–12]. Despite their irruptive behavior, Evening Grosbeaks also migrate on predictable schedules, especially during spring [13].

Migratory staging sites represent potential opportunities to monitor populations as aggregations occur at rich feeding areas. Each spring, large flocks of Evening Grosbeaks assemble at a migratory stop-over site, the campus of Oregon State University (OSU) in Corvallis, Oregon. The grosbeaks are normally present for a few weeks in April and May when elm (*Ulmus* spp.) seeds are their primary food source. Numerous, colorful and clamorous, the grosbeaks draw public attention. Newspaper reports from the 1970s indicated the number of birds was sometimes so large that students walking between classroom buildings carried umbrellas, even on sunny days, to shield themselves from grosbeak droppings (Appendix A). With tree canopies full of grosbeaks, many birds even descended to the ground to eat fallen elm seeds. Impressed by the large numbers of grosbeaks, statistics professor Fred Ramsey used a randomized sampling strategy to count birds in selected elm trees. He estimated 150,000 to 250,000 birds were foraging on campus one spring day in the mid-1970s (F. Ramsey, pers. comm.).

Our objectives were to count Evening Grosbeaks during their spring migratory stop-over in Corvallis, Oregon, and to compare our counts with Ramsey's estimate. We anticipated numbers were likely to be lower than previous estimates, so we conducted exhaustive area searches of the campus and counted all grosbeaks we encountered. We evaluated possible explanations that could influence local, regional and range-wide changes in abundance. Our surveys also established benchmark measurements of the number of birds with which future comparisons may be made.

## 2. Materials and Methods

**Study Area**. We studied Evening Grosbeaks on the campus of Oregon State University in Corvallis, Oregon. Evening Grosbeaks arrive as migrants, typically beginning in early April, to feed on elm seeds. Elms were planted across campus, with the first 35 being planted in 1913. Elms increased in number to a maximum of 332 mature trees and were the dominant canopy tree on campus by 1978. That year, Dutch elm disease was spreading across North America. Concerns that the arrival of the disease would cause widespread death of the elms, leaving the campus barren of shade trees, led to the removal of elm trees to ensure that root connections between infected trees would not allow the spread of the disease too quickly. Removal was spread across 10 years, with each set of removed trees being replaced with resistant elm varieties or other tree species. Today, 143 mature elms grow on campus.

**Counting methods**. We counted grosbeaks using an area search approach that covered the main Oregon State University campus containing elm trees (~170 ha). Elm trees grow in rows alongside walking paths across campus. We divided the campus into 4 zones (40–50 ha each), which were covered in an approximately equal number of visits each spring. We did not use the same stratified random selection of elm trees originally implemented by Ramsey because our preliminary observations in 2012 showed that nearly all trees had zero grosbeaks. Trained observers typically worked in pairs or small teams of up to 4 individuals. We walked through each zone for 30 to 45 min listening for and observing grosbeaks. When grosbeaks were detected, we counted them, or estimated their numbers to the nearest 10 individuals if the grosbeaks were flying or were partially obscured by canopy vegetation. We marked the locations of the birds on maps. If grosbeaks were detected while in flight, the direction headed by the flock was mapped, and time was noted to reduce chances of double-counting flocks that entered a neighboring zone during another active counting effort. Counts were conducted between 11 April and 31 May, 2013–2015. We typically counted grosbeaks between 7 a.m. and 1 p.m. Grosbeaks appeared to be roosting at night to the northwest of Corvallis in the Coast Range mountains. Flocks began arriving at the

study area from the northwest around 7 a.m. each day, then departed back to the northwest in late afternoon. We avoided counting grosbeaks on days with excessive rain or wind. Although elm trees were the focus of our surveys, we counted all grosbeaks regardless of the tree species in which we found them.

Grosbeaks are noisy and conspicuous birds during spring, which reduced concerns about detectability issues. The largest concern regarding error involved the accuracy of counts when grosbeaks occurred in flocks. Estimating flock sizes, especially of flying birds, required practice and consensus of the multiple observers in each survey group. When perched in tree canopies, especially after leaf-out later in spring, grosbeaks were sometimes obscured by foliage. Again, counts were achieved via consensus of multiple observers in each survey group. To help evaluate the accuracy of counts of perched flocks, we often waited for flocks to depart trees, so that all birds could be seen together at once as they passed overhead. We assume that counting errors were similar between eras, although we note that the historic count was estimated to be between 150,000 and 250,000.

**Data analysis**. We focused our comparison on the relative magnitude of change between the 1970s survey and the maximum numbers we detected during our 3-year study. Because the 1970s estimate is based on a single, unpublished assessment by a professional statistician, our comparison is simply one of relative scale of change. We report but did not statistically compare small variations in numbers among the 3 years of our own investigation. Given that we did not capture and mark the birds, we have no information on how long individual birds stayed during spring and no information on how much individuals moved around campus. Therefore, we report counts as the raw numbers found in each study zone as well as the maximum counts per week, which we calculated as the total number of birds reported during a week across all four geographic survey zones.

## 3. Results

We counted 8407 Evening Grosbeaks during 44 h and 13 min of effort distributed across 117 surveys, in 2013–2015 (Figure 1). Numbers peaked between 23 April and 2 May. Lowest numbers were counted in 2013 when, during 18 surveys (10 h 37 min), we found only 225 total grosbeaks. The spring of 2014 produced 7682 birds across 81 surveys (25 h 15 min); the maximum single-day count was 1442 on 2 May (Figure 2). Numbers were lower in 2015 when we conducted 18 surveys (8 h 21 min) and recorded 500 grosbeaks. Surveys in 2013 and 2015 detected maxima of 56 (2 May 2013) and 289 (23 April 2015) grosbeaks.

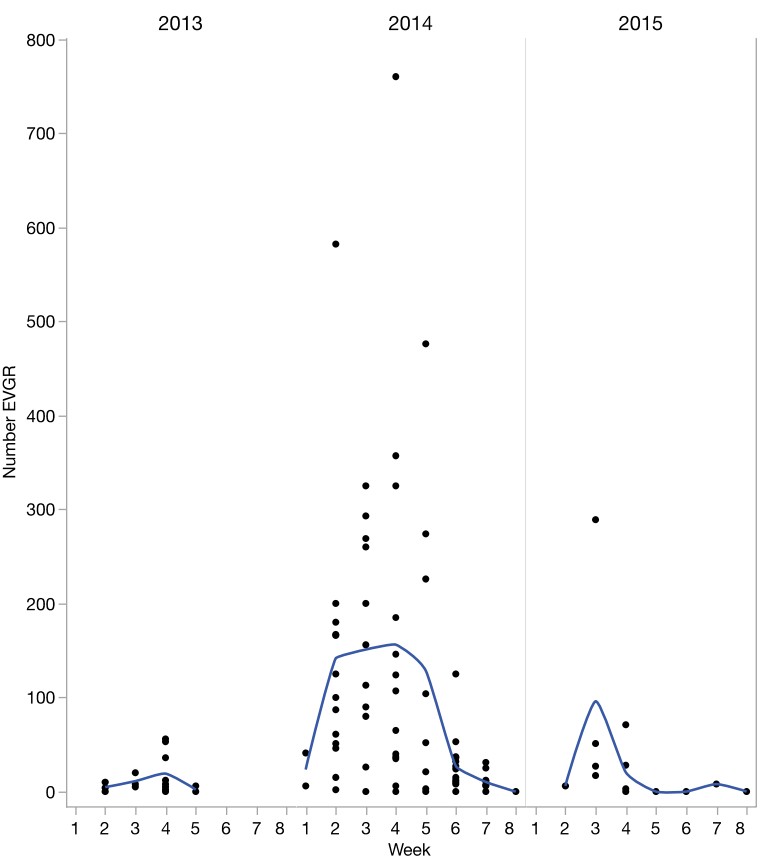

**Figure 1.** Numbers of Evening Grosbeaks detected during each survey of four sampling zones on the Oregon State University, Corvallis, campus, 2013–2015. Surveys began each year on 11 April (week 1) and extended for 8 weeks through May. Lines are Savitzky–Golay-fitted means [14] implemented with JMP [15].

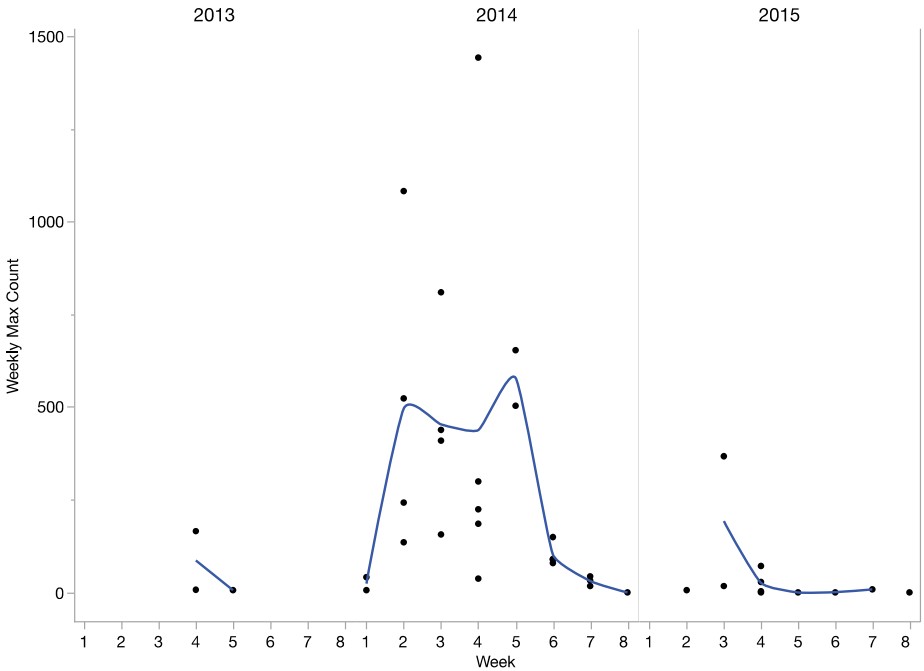

**Figure 2.** Maximum daily counts aggregated by week, 2013–2015, on the campus of Oregon State University, Corvallis, Oregon USA. Counts of zero are not illustrated (see Figure 1). Lines are Savitzky–Golay-fitted means [14] implemented with JMP [15].

## 4. Discussion

We found far fewer Evening Grosbeaks at this spring migratory stopover site than the very large numbers reported to have occurred during the 1970s. Although the original estimate of a maximum of 250,000 birds was produced by a professional statistician, the estimate was not published in peer-reviewed literature. Anecdotal accounts in newspapers corroborated the observation that grosbeaks were abundant. Nevertheless, even if the original estimate were an order of magnitude too high, it seems safe to conclude that Evening Grosbeak numbers are substantially lower than four decades ago at our study site. Our highest daily count was 1442 birds. The highest count reported in eBird from elsewhere in Corvallis was 1830 on 24 April 2014. We estimated that numbers peaked in Corvallis at a few thousand birds in just one (2014) of the three springs. Although we discontinued our standardized searches after 2015, Evening Grosbeaks are regularly reported in eBird from our study area. Between 2016 and 2022, the highest count reported was 500 birds (in 2022), suggesting that the three years of our study did not occur during a period of atypically low numbers by current standards. During the last decade, instead of grosbeaks flushing from the ground like wind-blown leaves as pedestrians cross campus, piles of elm seeds accumulate uneaten. The food is still plentiful, but the birds are not. We discuss some possible explanations as well as limitations of our interpretation.

We cannot definitively exclude counting error as a possible explanation because no further details were preserved from the statistician's estimate. Yet, the magnitude of the difference between the 1970s estimate and our 2013–2015 counts is so great that large errors, in either era, would not negate the primary conclusion of substantial declines in numbers of Evening Grosbeaks at this spring migration stop-over site. Newspaper articles in the 1970s reported regularly on arrivals of the migratory grosbeaks (analogous to the swallows of Capistrano), their exceptional abundance, and concerns regarding the effects of elm removal on grosbeak numbers. Grosbeaks were noted by a wildlife professor quoted in one newspaper article as being "somewhat erratic in their numbers", suggesting that the count noted by statistics professor Ramsey might be considered an estimate from one of the years with highest numbers. Evening Grosbeak abundances have been long known to be quite annually variable [6,8], but data on the number of years population booms continue at specific locations appear limited or absent.

Were grosbeak numbers influenced by changes in the availability of elm trees? Elm trees are still tall, mature and producing abundant seeds in our study area. The total number of elms on campus was reduced by more than 50% because of removals associated with Dutch elm disease and the replanting of resistant varieties and species. Current elms are canopy trees approximately 25 to 35 m tall on average and produce abundant seeds each spring. We are unaware of any data indicating that elm disease-resistant varieties may produce food that is less plentiful or less favored by grosbeaks but recognize the nutritional value of tree seeds does vary and regard this as an untested hypothesis [16,17].

We see no evidence for a phenological mismatch in the production of elm seeds with the timing of migration of Evening Grosbeaks. Our observations indicate that grosbeaks typically consume the fresh, green seeds, mostly while the seeds are still attached to trees. Seeds were plentiful while the highest numbers of grosbeaks were present in Corvallis in all 3 years of our study. Anecdotally, we noted large accumulations of uneaten seeds on the ground below elm trees in all years of our study. Newspaper and other anecdotal reports from the 1970s indicated that Evening Grosbeaks commonly descended to the ground to consume fallen seeds. We never observed grosbeaks on the ground. We consider it unlikely that the lack of availability of elm seeds explains the declines in grosbeak numbers.

We suggest it is unlikely that Evening Grosbeaks simply shifted their foraging sites to other nearby parts of Corvallis or the greater Willamette Valley. We inspected eBird data (2004 to 2021) to determine if higher numbers of Evening Grosbeaks were consistently reported elsewhere in Corvallis. A few higher counts than those we observed on campus were reported around Corvallis, but the maximum single-day count was less than 2000 birds. Most high counts were between 90 and 500 birds. Similarly small counts were reported

across the entire Willamette Valley. No counts were within even an order of magnitude of the 150,000–250,000 estimated in Corvallis in the 1970s. We found no other spring counts in eBird for our study site before 1980. The only checklist from OSU in that era indicated Evening Grosbeaks were observed on 15 April 1976 with the note "thousands on campus". We conclude that the available evidence and our own experiences do not support the idea that Evening Grosbeaks have simply shifted their areas of activity away from the Oregon State University campus. In fact, the magnitude of declines we observed align well with declines quantified by the North American Breeding Bird Survey (BBS) data, which monitored birds along roadside routes since 1966. When we compare the lower value from Ramsey's original estimate (150,000) with our maximum daily count of 1442, we found an average annual decline of −2.6%. Declines across the entire range of Evening Grosbeaks as quantified by BBS data show a decline of −2.5% [18]. Evening Grosbeak populations appear to be divided into five approximately geographically distinct groups, with all of the birds at our study site being members of the Type 1 population [19]. The breeding range for Type 1 is incompletely known, but current eBird data show it primarily occupies Oregon, Washington and British Columbia. BBS data indicate average annual abundance changes of −2.4% (Oregon), −1.8% (Washington) and −2.1% (British Columbia). The concordance of our observed decline with the BBS data suggests that reductions at our spring staging site reflect processes occurring across a much wider geographic extent.

The BBS trends for Evening Grosbeaks nationally and within the range of Type 1 are categorized as reasonably credible [18]. We noted that BBS data from Oregon included an average of four Evening Grosbeaks per route (N = 88) in the 1970s and less than 0.5 per route in the 2010s. An advantage of monitoring populations at stopover sites is the aggregation of large numbers of birds that will later disperse across vast geographic areas to breed [20]. Monitoring diffuse densities across large spatial extents poses greater difficulties tracking changes in abundance. In this regard, stopover sites such as ours can play useful roles for monitoring regional population changes in the same sense that hawk migration sites are useful for tracking large-scale variation in raptor numbers [21]. The average annual decline across four decades at our study site is similar to those found by the BBS, suggesting that large-scale processes may be responsible for the dramatic declines in abundance since the 1970s. Several possible explanations exist, all of which need careful evaluation with additional data.

As with most bird species, parasite and disease dynamics are poorly understood, yet are likely to be important drivers of population dynamics [22]. Evening Grosbeaks are known to be affected by several disease-causing organisms [23–26], but no assessments of these effects on demography have been conducted. Evidence for the influence of land-use and climate change on bird populations, in contrast, is substantial. For breeding populations of Evening Grosbeaks, changes in tree harvesting practices could influence the age structure of forests, altering the availability of primary foods such as spruce budworms or conifer seeds [5]. For example, coniferous tree species vary in their susceptibility to spruce budworm outbreaks [27,28]. To reduce tree mortality associated with defoliation by caterpillars, one strategy used by forest managers has been to re-plant after outbreak events and subsequent salvage operations with tree species less susceptible to spruce budworms [29,30]. The strategy likely decreases the overall availability of food for grosbeaks and other species that utilize spruce budworms, limiting the degree to which bird populations may soar during and immediately after outbreaks. Thus, we might expect that, over time, as a result of shifts in tree community composition, the probability of seeing very high abundance peaks of grosbeaks would decline. Furthermore, aerial application of insecticides across vast areas of forest to control outbreaks might also influence the magnitude of population increases in grosbeaks, but this needs further study [31,32].

The timing of spruce budworm outbreaks is also relevant to consider. Ramsey's high count of grosbeaks in Corvallis in the 1970s coincided with some of the largest spruce budworm outbreaks on record [10,33]. Although data are sparse for western North America, an outbreak beginning around 1965 and extending for two decades covered

at least 55 million hectares of eastern Canada. The fraction of CBCs reporting Evening Grosbeaks in the midwestern United States reached its peak then as well, with more than 90% of counts detecting grosbeaks compared with less than 10% in the early 1940s [10]. Documented budworm outbreaks in eastern North America have occurred for hundreds of years, as judged by tree-ring evidence in white spruce (*Picea glauca*) [27]. Within the last two centuries, major outbreaks occurred starting in 1877, 1910, 1945 and 1965 [33]. The abundance of Evening Grosbeaks on Christmas Bird Counts, especially during the 1970s and 1980s, strongly suggests that continental population sizes were very high and associated with the budworm outbreak. The declines since then have been steep and very apparent in CBC data [4,34].

Evening Grosbeaks are known to have four distinct 'call types' that appear to align with geographically distinct populations in the United States and Canada [19]. All Evening Grosbeaks we detected during spring migration in our region of western Oregon are type 1, which breeds from southern Oregon north to central British Columbia. The extent to which call type 1 wanders across North America and mixes with other call types remains unclear, but records validated by sound recording extend as far east as Colorado [19]. Interestingly, the apparent peak in numbers at our study site in the 1970s does not align well with documentation of western spruce budworm outbreaks in British Columbia. From 1909 to 1950, five short-lived (2–8 years each) western spruce budworm outbreaks occurred, followed by 48 years of outbreaks from 1967 to 2014 [35]. Although outbreaks in the west appear to cover smaller spatial extents than those further east in Canada and the northeastern United States, the observation that outbreaks have been continuous for 55 years and increasing in geographic extent does not align well with the dramatic declines in Evening Grosbeak numbers. We are unaware of any similar datasets from Washington and Oregon. The degree to which western populations of Evening Grosbeaks are influenced by spruce budworm outbreaks needs additional study.

Numerical responses of bird populations to such variable food resources adds to the challenges of understanding population dynamics. Such fluctuations affect our efforts to evaluate and prioritize conservation options as well [36]. For example, widely reporting declines as the difference between the highest peaks and the lowest troughs of inherently cyclical or variable populations risks misrepresenting conservation urgency. Comparing Ramsey's estimate (a time period very likely to have been one in which Evening Grosbeak populations were exceptionally high) with our estimate (from a time period during which numbers may be very low) could exaggerate declines that might instead be components of long-term population cycles or, at least, fluctuations. The interpretation of population declines is also directly affected by the length of periodicity in population cycles, especially if that periodicity exceeds the duration of our historical data records. Christmas Bird Count data, for example, seem to reveal a long-term rise-and-fall in numbers that extends over a century, not a few decades [10]. Thus, a wider perspective on long-term change might influence how we perceive declines since the 1970s and how we prioritize conservation efforts [37]. The need for biodiversity count data, especially from planned surveys, to create reliable benchmarks against which future resurveys may be compared is essential [38]. In addition, the selection of study sites for tracking long-term changes in abundance must utilize carefully considered designs to minimize bias that may produce misleading results [39,40]. Furthermore, increased perspective must also be weighed against the axiom in population biology theory that populations fluctuating the most strongly are at greater risk of extinction than less variable populations [41]. Species with population dynamics like Evening Grosbeaks warrant closer monitoring and careful evaluation of the factors influencing their population fluctuations.

In summary, we conducted a snapshot comparison of two points in time, the 1970s and 2013–2015, to produce an estimate of change in Evening Grosbeak numbers at a migratory stopover site. Even if the original maximum estimate of 250,000 birds is associated with an unknown level of uncertainty, our observations of maximum counts of less than 5000 birds at the same site provides clear qualitative evidence that declines have been large, even if

the statistical precision of the estimates is unavailable or highly uncertain. Our results align with BBS data from across North America and within the range of Type 1 Evening Grosbeaks that numbers have been reduced substantially since the 1970s [4,5,10]. Additional perspectives on factors associated with long-term population fluctuations and declines are needed. In particular, planned surveys designed to take advantage of opportunities presented by spring stopover sites will improve our understanding of population fluctuations through time. Our observations suggest that further attention should be directed toward risk analyses and hypothesized explanations for the decline of this charismatic species.

**Author Contributions:** Conceptualization, W.D.R.; methodology, W.D.R., J.G., J.M., T.A.H. and J.R.C.; data collection, W.D.R., J.G., J.M., T.A.H. and J.R.C.; formal analysis, W.D.R.; writing—original draft preparation, W.D.R.; writing—review and editing, W.D.R., J.G., J.M., T.A.H. and J.R.C.; project administration, W.D.R.; funding acquisition, W.D.R. All authors have read and agreed to the published version of the manuscript.

**Funding:** This research was supported by the Bob and Phyllis Mace Professorship to W.D.R.

**Institutional Review Board Statement:** Not applicable.

**Informed Consent Statement:** Not applicable.

**Data Availability Statement:** The raw data are reported in Figures 1 and 2.

**Acknowledgments:** Fred Ramsey helped place our surveys in an historical context. Many observers contributed to the study, including the OSU Bird Nerds, who embraced the Gobs of Grosbeaks project. Lok Yung Chen researched historical reports of grosbeaks in local newspaper archives. WDR was supported by the Bob and Phyllis Mace. Thank you to Michael Brawner, Will Hemstrom, Adrian Hinkle, Christopher Hinkle, Ed Jenson, Nathan Jones, Emily Love, Travis MacDonald, David Mellinger, Noelle Moen, Kim Nelson, Emily Platt, Melanie Ripley, Dan Roby, Jennifer Rose, Josée Rousseau, Ed Seusen, Ian Souza-Cole, David Vasquez, Hayati Woolfenden, and other ornithology students. We thank the Evening Grosbeak Working Group, especially Thomas Hahn, Aaron Haiman, John Woods and Matt Young, for feedback that improved the paper.

**Conflicts of Interest:** The authors declare no conflict of interest.

## Appendix A

Poem published in the Corvallis Gazette-Times, 1 May 1978, by T. N. Tillman.
Hey had you noticed? The grosbeaks are back.
Those birds that are yellow, white and black.
They are chirping and eating up high in the trees.
And releasing their droppings with such affluence and ease.
It is literally raining, raining bird-do.
That makes splotches of white on me and on you.
You say birds of rare beauty? You're really quite right, sir.
But who of us needs all that airborne fertilizer?
As for me and those birds called evening grosbeaks,
I just wish that someone could stop up their leaks.

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
