# Peer review of "Dramatic Declines of Evening Grosbeak Numbers at a Spring Migration Stop-Over Site"

_diversity, doi:10.3390/d14060496_

Round 1

Reviewer 1 Report

Robinson et al. present an interesting case study examining change in abundance of evening grosbeaks at on location in Oregon, USA spanning a timeframe of over 40 years. The article is well written and, given the limitations of the data used and very small sample in space, I appreciate the care with which they presented limitations to interpretation of these results. Nevertheless, such a substantial decline in numbers of evening grosbeaks is striking and matches what could be inferred from other studies (mostly from the eastern U.S.A.).

I do think this paper can and should be strengthened by doing their own analysis of existing data sets from large-scale sampling (e.g. Christmas Bird Count and eBird) for the Pacific Northwest. Inferring population declines as opposed to range shifts in response to climate, food availability, habitat changes, or other causes is much weaker without at least looking at regional trends across the western U.S. 

Some related comments from specific sections include:

Lines 52-54: A citation for the existence and consistency of regular spring migratory stop-overs in Evening Grosbeaks is needed here. As an irruptive species, this appears to be a rare - if it exists at all - phenomenon. References of newspaper reports from the 1970s indicating larger numbers in spring is not sufficient to establish this as a regular spring migration preparation site, especially given that the “migratory” patterns of grosbeaks are likely to be semi-irregular at least. This is also critical for any inferences made about trends observed at this one site.

Lines 198-212: yes it may appear that grosbeaks have not shifted their area of activity within the Corvallis region or Willamette Valley. But, given the likely magnitude of movements for these birds in response to food abundance, it is possible they’ve shifted at a continental scale. The lack of examination of trends using eBird data at a larger scale (which would be easy to do) is a significant and easy to correct, hole in this analysis and interpretation

Lines 221-229: Loss of spruce budworm is possible, but we also know that budworm outbreaks in the pacific northwest have been spreading and are more persistent in a warming climate, suggesting that despite forest management there could be abundant food causing shifts in spatial abundance. Further review and citation of what recent patterns of budworm outbreaks and their spatial extent would make this stronger.

Line 272: Yes these trends could be indication of long-term fluctuations. But, it could also be range shift which it would be good to attempt to address this with your own evaluation, summary and presentation of CBC and eBird data for the west.

Author Response

Robinson et al. present an interesting case study examining change in abundance of evening grosbeaks at on location in Oregon, USA spanning a timeframe of over 40 years. The article is well written and, given the limitations of the data used and very small sample in space, I appreciate the care with which they presented limitations to interpretation of these results. Nevertheless, such a substantial decline in numbers of evening grosbeaks is striking and matches what could be inferred from other studies (mostly from the eastern U.S.A.).

I do think this paper can and should be strengthened by doing their own analysis of existing data sets from large-scale sampling (e.g. Christmas Bird Count and eBird) for the Pacific Northwest. Inferring population declines as opposed to range shifts in response to climate, food availability, habitat changes, or other causes is much weaker without at least looking at regional trends across the western U.S. 

--Thank you for the feedback. We appreciate that you noticed we were trying to be exceptionally careful and not over-interpret the data. As you noted, the historic data point is anecdotal and the exact details of how it was calculated have been lost. Despite that shortcoming, additional anecdotal information from newspaper accounts does add some general level of ‘validation’ given that those accounts do not sound anything like what people would report today. The grosbeaks are hardly even noticed by the public today and we did exhaustive area searches to find them; no need to randomly choose trees like Ramsey did. Thus, it’s an interesting case study, but also needs to be handled carefully so as to learn from it but not get too carried away. We saw the interpretation of the results as a way to explore possible explanations and also to mention some challenges of comparing snapshot samples of bird numbers across time.

We added new content after we studied the BBS data. The eBird and CBC data present substantial challenges (as explained below), so we focused on BBS. The average annual decline we noted (-2.6%) when comparing Ramsey’s estimate with ours is very similar to the range-wide decline (US and Canada) documented by BBS results (-2.5%). It is also very similar to the decline specific to Oregon (-2.4%). It appears, then, that the decline we noted is concordant with declines at larger scales so this at least gives us a little confidence that our local case study is not undermined by the possibility that grosbeaks have simply moved somewhere else to stage during spring.

Winter counts from CBCs might also give general ideas of changes, but there are many problems with CBC data, as the reviewer undoubtedly knows. Among the unique challenges presented by Evening Grosbeaks is that they appear to have 5 different types (based on distinctive calls [one is in Mexico, which we ignore here]) and we are not sure where the type 1 birds that stop in Corvallis during spring spend the winter. As far as we know, there is no information on how many types might over-winter together nor is there any information about the degree of site fidelity each type has from winter to winter. So a region-wide analysis of CBC data probably will not help much because it is not clear what the limits to the region should be. Over time, as more eBirders load recordings with their grosbeak observations, we will be able to refine our knowledge of the range limits of each type for the entire annual cycle. The existing information from CBCs, which we reference in the introduction and later in the Discussion, aligns with our observations of higher numbers in the 70s and much lower numbers now. Perhaps the most reliable data are from Project Feederwatch, which uses standardized protocols and contains 10,000 sites. The data in Bonter and Harvey span 1988-2006, so start about 10 years after Ramsey’s estimate. Mean flock sizes at feeders across the US and Canada dropped from 12 to 9 during the 30 year time frame and the percent of sites with grosbeaks dropped from 0.19 to 0.1. In the Pacific Northwest, data interpolation indicates that the proportion of feeders with grosbeaks during winter dropped 30% from the early 1990s to early 2000s. Again, we didn’t add much about winter to the revised paper because we don’t know which type is being observed at the feeders or on CBCs.

The major shortcoming of eBird as a tool for corroborating our result is that it generally lacks data from the 1970s so we cannot use it as an independent dataset to corroborate the pattern. It really took off as a popular bird count archival tool this century, especially since 2010. So there’s no information from the PNW that will help us see changes across 50 years. Even if it were available, there might be some hesitation about trusting numbers, especially large counts, reported there. We recently conducted a study that showed counting errors are probably quite large, even in situations where detection is not an issue, especially for very common birds (https://www.frontiersin.org/articles/10.3389/fevo.2021.568278/full). We did find one checklist from OSU campus in April 1978 that reported grosbeaks in the “thousands” but did not give a better estimate.

In the end, despite the anecdotal nature of the original count, we feel the title of the paper is accurate and it is safe to conclude that the birds have declined at THIS spring stop-over site. It appears that the magnitude of the decline in Corvallis currently aligns with trends at larger scales, but of course, this all needs additional study, which we call for in the paper.

We have tried to revise certain portions of the paper in response to ideas from both reviewers to clarify our thoughts. See track changes for those revisions. Thank you for being thoughtful, polite and sharing our concerns to carefully interpret the case study results.

Some related comments from specific sections include:

Lines 52-54: A citation for the existence and consistency of regular spring migratory stop-overs in Evening Grosbeaks is needed here. As an irruptive species, this appears to be a rare - if it exists at all - phenomenon. References of newspaper reports from the 1970s indicating larger numbers in spring is not sufficient to establish this as a regular spring migration preparation site, especially given that the “migratory” patterns of grosbeaks are likely to be semi-irregular at least. This is also critical for any inferences made about trends observed at this one site.

--We understand the concern here. From our experience over the last 15 years, the grosbeaks are on a predictable migratory schedule, arriving at the same time each spring and departing for breeding sites by end of May. So, although viewed from broad perspective, they are considered to be irruptive, they also have regular migration schedules, at least in the PNW. They are here every year, not missing any years. We’ve added a general reference about their migratory status here in Oregon around line 56.

Lines 198-212: yes it may appear that grosbeaks have not shifted their area of activity within the Corvallis region or Willamette Valley. But, given the likely magnitude of movements for these birds in response to food abundance, it is possible they’ve shifted at a continental scale. The lack of examination of trends using eBird data at a larger scale (which would be easy to do) is a significant and easy to correct, hole in this analysis and interpretation

--We don’t think the shift is happening at a continental scale because the 5 types tend to occupy mostly different geographic regions in spring and summer. The distribution and movements of the call types still requires additional study, especially regarding how much they might mix outside of spring and summer. But in field work here over the last 15 years, we have never detected any other call type but type 1. So, it’s highly unlikely they are redistributing at a continental scale. That being said, they could possibly redistribute elsewhere in the PNW, which we have explored to some extent (mostly looking for other sites across northwest Oregon). Again, with respect to eBird, the data do not extend back in time far enough to do a reasonable evaluation of trends across time and space.

The closest approximation we can get is BBS data, which do not address the migratory period (the earliest run routes each year start when the EVGR are typically done migrating). As we say in the revised Discussion, the results from BBS appear to be quite similar to our own observations.

Are there other ways that 150,000 birds could be redistributed? Sure. But with the current coverage by birders it is almost certainly true that any giant aggregation that shifted somewhere else would have been found. We know of no evidence for such an aggregation. Is it technically possible that 150,000 birds could simply be thinly distributed across a much larger area and still be present? Yes. The trouble is, even if we analyzed all recent EVGR counts and built a reliable species distribution and abundance map to then estimate total population sizes, we don’t have any data from 1970s to build analogous maps and generate similarly reliable abundance estimates. That’s why we call for implementation of more standardized surveys of this interesting species.

Lines 221-229: Loss of spruce budworm is possible, but we also know that budworm outbreaks in the pacific northwest have been spreading and are more persistent in a warming climate, suggesting that despite forest management there could be abundant food causing shifts in spatial abundance. Further review and citation of what recent patterns of budworm outbreaks and their spatial extent would make this stronger.

--We are unaware of the evidence that the PNW is warming enough to cause more and longer budworm outbreaks. The evidence we have seen is that temperature increases in the PNW over the last 50 years are minimal, averaging around 0.5C or less normally. We do think it is reasonable to pose the possibility that changes in spruce budworm abundance have an effect on the overall population sizes of EVGR, a possibility we raise but do not get too carried with (we hope; that was our intent anyway). If the reviewer knows of key literature we have overlooked about budworms in the west, we would be interested in learning more.

Line 272: Yes these trends could be indication of long-term fluctuations. But, it could also be range shift which it would be good to attempt to address this with your own evaluation, summary and presentation of CBC and eBird data for the west.

--We have clarified what the CBC data show. As we mentioned before, eBird data don’t help us because of the limited temporal scope they currently have. All the better to argue that more rigorously accurate counts are needed to be added now to eBird so that others decades from now can have more reliable data than we have at our own disposal currently.

I think, in the end, the reviewer is concerned about the link between local stop-over site changes and the possibility that birds have been redistributed elsewhere. We don’t have a definitive way to address that possibility, which we try to say in the Discussion. Summaries of other available data all point to the conclusion that numbers were higher in the 1970s than they are today. We think the current paper is best used to report our observations and suggest hypotheses that can guide, we hope, the future implementation of better data collection practices.

Thank you very much for reading and thinking about the paper.

Reviewer 2 Report

This is an unusual study, in that there is not any probabilistic depiction of uncertainty in the key inferences. I think I can accept that the recent counts provide some indication of potential local grosbeak decline on OSU's campus that perhaps warrants further rigorous study. It's less clear to make of these findings big-picture. The authors spend a fair amount of time describing/discussing broader implications. While I thought the authors generally were careful about depicting uncertainty at the broad scale, my sense is that the primary scope of the study is really this locality, so I'd narrow the focus a bit.

Obviously, the elephant in the room--which the authors do acknowledge up front--is that there's little information about how the estimate from the 1970's was generated and consequently little way for a reviewer to gauge the strength of evidence here. It would be helpful to have more detail here. Still, perhaps it is still ethically reasonable to present this as (largely) anecdotal evidence for a local decline that warrants subsequent attention and follow-up.

L15: Wasn’t quite sure what this sentence meant.

L51: All densities vary through space and time—do the authors mean “vary dramatically” [substantially]?

L61: Seems like it would be useful to have more details about the procedure beyond the randomized sampling strategy, although I recognize this may not be available. 

L104: Presumably, even if detection is imperfect, it was also imperfect during the 1970’s too…could note this.

L113: It’s a little unusual for a scientific manuscript (at least, one that it is not strictly descriptive) to lack any statistical analysis, but I think that’s ok here—it would be difficult to craft an analysis that addressed the uncertainty in a satisfactory way.

L196: Is it possible that this decline is really a decline from anomalously high number during the historic period—e.g., grosbeak were so surprisingly abundant that there was some impetus to count them and write about them---vs. evidence for some broader trend? Or is it possible evidence for decline arises from site-selection bias (doi.org/10.1111/cobi.13371)?

Discussion: to me, the discussion runs a little long. Partially this is because the authors are providing caveats and trying to place the results in context with alternative/existing research and evidence, which is completely understandable and appropriate. I think it’s reasonable to mention potential reasons for decline (eg., food, parasites, etc.) relative to the late 1970’s, but I might keep this more succinct—much of this part of the discussion is somewhat vague and speculative (understandable given that broad scale population dynamics are difficult to study, but as a whole, perhaps a little beyond the scope of the study).

Maybe this is encapsulated in the paragraph at L264. On L209, the authors note that the local declines they observe strongly suggest large regional and perhaps range-wide declines, which implies this is a conservation concern. At L264 and subsequently, the authors pivot to a broader view—that there are some broader rise and fall trends going back over the century (i.e., maybe any decline is not necessarily a conservation concern, but a return to a longer-term baseline condition, or some natural/slow cyclicity). To me, the difficulty relates to cross-walking the local evidence with broader regional/temporal narratives that are shrouded in more uncertainty (again, because understanding dynamics at these scales is hard).  I can more or less “buy” that there is some evidence for local decline that deserves further research and scrutiny, but it’s not clear to me that this potential local decline can be  generalized to make statements about broader grosbeak status and trends in a clear way. 

Author Response

This is an unusual study, in that there is not any probabilistic depiction of uncertainty in the key inferences. I think I can accept that the recent counts provide some indication of potential local grosbeak decline on OSU's campus that perhaps warrants further rigorous study. It's less clear to make of these findings big-picture. The authors spend a fair amount of time describing/discussing broader implications. While I thought the authors generally were careful about depicting uncertainty at the broad scale, my sense is that the primary scope of the study is really this locality, so I'd narrow the focus a bit.

Obviously, the elephant in the room--which the authors do acknowledge up front--is that there's little information about how the estimate from the 1970's was generated and consequently little way for a reviewer to gauge the strength of evidence here. It would be helpful to have more detail here. Still, perhaps it is still ethically reasonable to present this as (largely) anecdotal evidence for a local decline that warrants subsequent attention and follow-up.

--Thank you for noticing that we trying to be careful and not over-extend. It is an interesting data point from the 1970s but certainly not one that had any additional information saved with it, unfortunately. Nevertheless, despite that shortcoming, additional anecdotal information from newspaper accounts does add some level of ‘validation’ given that those accounts do not sound anything like what people would report today. The grosbeaks are hardly even noticed by the public today. Thus, it’s an interesting case study, but also needs to be handled carefully so as to learn from it but not get too carried away. We saw the interpretation of the results as a way to explore possible explanations and also to mention some challenges of comparing snapshot sample of bird numbers across time. It’s not an easy task!

We agree with your careful interpretation. We can learn from this possible pattern of decline and suggest follow-ups, trying to understand the range of possible causes for a local change in numbers. Hopefully the exploration of ideas will guide other comparisons with historic snapshots, if you will, as well.

L15: Wasn’t quite sure what this sentence meant.

--We removed the sentence.

L51: All densities vary through space and time—do the authors mean “vary dramatically” [substantially]?

--Thank you.

L61: Seems like it would be useful to have more details about the procedure beyond the randomized sampling strategy, although I recognize this may not be available. 

--Unfortunately, the details were not saved by the statistician. We agree, of course, that this is a shortcoming of the data point. Had we been unable to find other information, such as the newspaper articles, to offer some degree of corroboration that EVGR were extremely numerous, we would have let the data slip into the ether forever. But the articles all seem to suggest that the birds were so common they were hard to miss. That’s hardly the case at all today. Most people on campus have probably never noticed them because they are so uncommon they don’t forage on the ground any longer nor do they defecate on people constantly as the people walk across campus.

L104: Presumably, even if detection is imperfect, it was also imperfect during the 1970’s too…could note this.

--We added sentence to this paragraph stating that we assume counting errors to be similar between eras.

L113: It’s a little unusual for a scientific manuscript (at least, one that it is not strictly descriptive) to lack any statistical analysis, but I think that’s ok here—it would be difficult to craft an analysis that addressed the uncertainty in a satisfactory way.

--Agreed.

L196: Is it possible that this decline is really a decline from anomalously high number during the historic period—e.g., grosbeak were so surprisingly abundant that there was some impetus to count them and write about them---vs. evidence for some broader trend? Or is it possible evidence for decline arises from site-selection bias (doi.org/10.1111/cobi.13371)?

--We really appreciated the article that was referenced here. It’s almost certainly the case that the count was made by statistician Ramsey because he noticed the birds were so numerous and wanted to know how many there were. Without continuous long-term data, we simply don’t know for sure; that’s always the challenge of interpreting comparisons with historical data. That being said, we have tried to stick to what we think is the most reasonable conclusion, that grosbeaks HAVE declined in numbers when comparing his estimate with ours, and then we try to explore the possibilities associated with potential broader scale declines and hypothesized causes that might “trickle down” and influence what we see at a local site. Certainly, the CBC data (and the Project Feederwatch data) as well as BBS data (which we added information from in the Discussion) indicate there have been broad-scale declines consistent with our observations at this one location. We’ve revised some of the text to make clearer the evidence from BBS data, as this was also requested by the other reviewer. Please see tracked changes.

Discussion: to me, the discussion runs a little long. Partially this is because the authors are providing caveats and trying to place the results in context with alternative/existing research and evidence, which is completely understandable and appropriate. I think it’s reasonable to mention potential reasons for decline (eg., food, parasites, etc.) relative to the late 1970’s, but I might keep this more succinct—much of this part of the discussion is somewhat vague and speculative (understandable given that broad scale population dynamics are difficult to study, but as a whole, perhaps a little beyond the scope of the study).

--Thank you. Your interpretation of our intention is correct. We have re-read the discussion and made some edits to streamline the presentation. Tracked changes are preserved. We have tried to balance the feedback from you to trim this a bit with advice from friendly reviewers who most often said “why didn’t you write about topic X, too?”

Maybe this is encapsulated in the paragraph at L264. On L209, the authors note that the local declines they observe strongly suggest large regional and perhaps range-wide declines, which implies this is a conservation concern. At L264 and subsequently, the authors pivot to a broader view—that there are some broader rise and fall trends going back over the century (i.e., maybe any decline is not necessarily a conservation concern, but a return to a longer-term baseline condition, or some natural/slow cyclicity). To me, the difficulty relates to cross-walking the local evidence with broader regional/temporal narratives that are shrouded in more uncertainty (again, because understanding dynamics at these scales is hard).  I can more or less “buy” that there is some evidence for local decline that deserves further research and scrutiny, but it’s not clear to me that this potential local decline can be  generalized to make statements about broader grosbeak status and trends in a clear way. 

--Correct. Cross-walking is indeed the big issue. Were the declines happening across the range of type 1, or the West, across the whole continental range, or just at this one place? We have revisited some of the larger datasets across the range (mostly BBS) and attempted to clarify our presentation that suggests the declines were not exclusive to our study site.

Thank you for reading the paper and sharing your comments.

Round 2

Reviewer 2 Report

I appreciate the edits and responses to comments--I don't have any further suggestions.